# Antimicrobial Resistance, Biofilm Formation, and Virulence Determinants in *Enterococcus faecalis* Isolated from Cultured and Wild Fish

**DOI:** 10.3390/antibiotics12091375

**Published:** 2023-08-28

**Authors:** Md. Liton Rana, Zannatul Firdous, Farhana Binte Ferdous, Md. Ashek Ullah, Mahbubul Pratik Siddique, Md. Tanvir Rahman

**Affiliations:** Department of Microbiology and Hygiene, Faculty of Veterinary Science, Bangladesh Agricultural University, Mymensingh 2202, Bangladesh; liton.21110215@bau.edu.bd (M.L.R.); alaila796@gmail.com (Z.F.); farhanaferdous1501184@gmail.com (F.B.F.); ashek.21110216@bau.edu.bd (M.A.U.); mpsiddique@bau.edu.bd (M.P.S.)

**Keywords:** fish, biofilm, antibiotic resistance, MDR, *Enterococcus faecalis*

## Abstract

Fish has always been an integral part of Bengali cuisine and economy. Fish could also be a potential reservoir of pathogens. This study aimed to inquisite the distribution of virulence, biofilm formation, and antimicrobial resistance of *Enterococcus faecalis* isolated from wild and cultivated fish in Bangladesh. A total of 132 koi fish (*Anabas scandens*) and catfish (*Heteropneustes fossilis*) were collected from different markets in the Mymensingh district and analyzed to detect *E. faecalis*. *E. faecalis* was detected by conventional culture and polymerase chain reaction (PCR), followed by the detection of virulence genes by PCR. Antibiotic susceptibility was determined using the disk diffusion method, and biofilm-forming ability was investigated by crystal violet microtiter plate (CVMP) methods. A total of 47 wild and 40 cultured fish samples were confirmed positive for *E. faecalis* by PCR. The CVMP method revealed four per cent of isolates from cultured fish as strong biofilm formers, but no strong producers were found from the wild fish. In the PCR test, 45% of the isolates from the wild and cultivated fish samples were found to be positive for at least one biofilm-producing virulence gene, where *agg*, *ace*, *gelE*, *pil*, and *fsrC* genes were detected in 80, 95, 100, 93, and 100% of the isolates, respectively. Many of the isolates from both types of samples were multidrug resistant (MDR) (73% in local fish and 100% in cultured fish), with 100% resistance to erythromycin, linezolid, penicillin, and rifampicin in *E. faecalis* from cultured fish and 73.08, 69.23, 69.23, and 76.92%, respectively, in *E. faecalis* from wild fish. This study shows that *E. faecalis* from wild fish have a higher frequency of virulence genes and biofilm-forming genes than cultivated fish. However, compared to wild fish, cultured fish were found to carry *E. faecalis* that was more highly multidrug resistant. Present findings suggest that both wild and cultured fish could be potential sources for MDR *E. faecalis,* having potential public health implications.

## 1. Introduction

Gram-positive *Enterococcus faecalis* is a common inhabitant of the gastrointestinal tracts of people, animals, and birds. It is also present in various environmental sources, including soil, water, and sewage [1]. Despite being a commensal bacterium, *E. faecalis* can cause severe infections in people, such as bacteremia, endocarditis, and urinary system infections [2]. In recent years, there has been growing concern about the increasing prevalence of antibiotic-resistant strains of *E. faecalis* in both human and animal populations [3]. Enterococcus is the second leading bacteremia and endocarditis-causing bacteria in the USA hospital. In Italy, it is the third most important bacteria that causes urinary tract infection (UTI) and blood infection in hospitalized patients [4].

Biofilm is a sessile microbial community of cells that have become permanently bonded to a surface, an interface, or each other [5]. Biofilms are important in the pathogenesis of enterococcal infections. This is another characteristic of *E. faecalis* associated with the bacterium’s persistence in the environment and its ability to cause infections. The ability of bacteria to produce biofilm helps them to colonize an inert surface and endows the capability of adhesion to the host cells [6]. The majority of hospitalized infection-causing bacteria are associated with biofilm formation. It is associated with more than 80% of diseases caused by enterococci in the USA [7]. Several virulence genes in *E. faecalis* have been identified that are involved in biofilm formation and pathogenicity. These genes are critical in the bacterium’s ability to form biofilms and cause infections.

Antimicrobial resistance (AMR) is a major global health crisis that has jeopardized the effectiveness of antibiotics and the capacity to combat bacterial infections. The World Health Organization (WHO) has recognized AMR as one of the most pressing threats to global health, ranking it among the top ten [8]. It estimates that by 2050, AMR will be responsible for causing 10 million deaths annually, thereby outpacing cancer as a primary cause of mortality [9]. The transmission of antibiotic-resistant genes to humans has been related to the appearance and propagation of antibiotic-resistant bacteria in the environment, particularly in food-producing animals and their byproducts [10]. Antibiotic resistance in *E. faecalis* is primarily due to the acquisition of resistance genes via horizontal gene transfer [11]. *E. faecalis*, a commensal of the intestinal microbiota in humans and animals, is a notorious opportunistic pathogen that has increasingly gained attention due to its association with hospital-acquired infections and foodborne illnesses. Antibiotic use in both human and animal populations contributes significantly to the selection and spread of antibiotic-resistant strains [12]. In addition, *E. faecalis* can also produce virulence factors, which are associated with the ability of the bacterium to cause disease in humans [8]. *E. faecalis* is innately resistant to broad-spectrum antibiotics routinely used in hospitalized patients, including aminoglycosides, cephalosporins, and macrolides [13], thereby making these bacteria very difficult to treat.

Fish has always been an integral part of Bengali cuisine and economy. As a potential sector, the fish industry plays a tremendous role in the development of the economy of the country. The fisheries industry in Bangladesh is vastly diversified, with two categories: the inland fisheries sector and the marine sector [11]. The inland captured and inland cultured are the main subsectors of the inland category. Inland culture includes the beel, rivers, and other natural sites where fish can be found. The inland captured subsector includes ponds, hatcheries, bio-flock cultures, and other artificial sites for fish culture [14]. Fish and fish products are important food sources for human consumption and are also used in the aquaculture industry. The use of antibiotics in aquaculture has been associated with the emergence of resistance in many bacterial species, including *E. faecalis* [10]. In addition, the use of antibiotics in aquaculture has also been linked to the spread of antibiotic-resistant genes in the environment. These MDR pathogens can easily enter the food chain from fish, thereby putting the consumer’s health at risk. There is no data focusing on AMR, biofilm formation and virulence gene distribution in *E. faecalis* from Koi and catfish in Bangladesh. Therefore, the study aimed to understand the prevalence and antibiotic resistance pattern, virulence factors, and biofilm production in *E. faecalis* isolates present in wild and cultured fishes.

## 2. Results

### 2.1. Occurrence of E. faecalis

Out of 132 samples, 87 (66%; CI: 57.48–73.44%) were found positive for *E. faecalis* in conventional PCR test by targeting the most conserved gene (*ddl_E. faecalis_*) present in *E. faecalis* (Figure 1). Among them, 47 (54%; CI: 43.60–64.10%) were found positive for *E. faecalis* in wild fish and 40 (46%; CI: 35.89–56.39%) in cultured fish. The occurrence of *E. faecalis* in wild fish was significantly (*p* < 0.001) higher than in cultured fish (Figure 2). All the characteristics and relationships among the biofilm formation, virulence, and antibiotic-resistant patterns in the *E. faecalis* isolates from cultured and wild fish are shown in Figure 2, Appendix A.

### 2.2. Biofilm Formation in E. faecalis

The majority of the *E. faecalis* isolates from wild-type fish samples, 68% (32/47, CI: 53.83–79.60%) harbored biofilm in the CMVP test. On the other hand, 87% (35/40, CI: 73.88–94.54%) of the isolates from cultured fish produced biofilm in the test. The biofilm-forming *E. faecalis* isolates were significantly (*p* < 0.006) higher in the cultured fish than in the wild fish (Figure 2). The CMVT test revealed that 5% (2/38, CI: 0.96–17.70%) of *E. faecalis* from cultured fish produced strong biofilm, while no isolates found strong biofilm producers in the *E. faecalis* from wild fish. The occurrence of intermediate biofilm producers in both types of samples was high (wild: 68%, CI: 53.83–79.60%; cultured: 82%, CI: 68.05–91.25%) compared to non-biofilm producers (wild: 31%, CI: 20.39–46.16%; cultured: 12%, CI: 05.46–26.11%).

### 2.3. Presence of Virulence Genes in E. faecalis

All the (*n* = 87) PCR-positive isolates from both types of samples exhibited the presence of at least four tested virulence genes. *E. faecalis* isolates from wild fish harbored the maximum number of virulence genes *fsrB* and *sprE* (100%), followed by *agg* and *gelE* (78%), *pil* (91%), *fsrA* (80%), ace (68%), and *fsrC* (51%) (Figure 3).

Moreover, the highest number of virulence genes found in *E. faecalis* from cultured fish was *fsrB* (100%), followed by ace and *sprE* (95%), *fsrA* and *pil* (87%), *gelE* (92%), and *agg* (45%). The occurrence of the *cyl* gene was negative in both types of samples.

In wild fish, the bivariate analysis of virulence genes showed a significant correlation between *agg* and *ace* (ρ = 0.014), *ace* and *gelE* (ρ = 0.032), *gelE* and *fsrC* (ρ = 0.00), *gelE* and *pil* (ρ = 0.00), and *fsrC* and *pil* (ρ = 0.033) (Appendix A). However, the study found no significant associations between the virulence genes and the isolates’ different degrees of biofilm formation ability (Appendix A).

In the bivariate analysis, *E. faecalis* in cultured fish showed a significant correlation between the *fsrA* and *sprE* (ρ = 0.000) and *fsrC* and *pil* (ρ = 0.007) genes (Appendix A). In addition, the statistics revealed a significant association of the virulence gene *gelE* with the different degrees of biofilm-forming ability of *E. faecalis* in cultured fish (Appendix A).

### 2.4. Antibiogram Profile of E. faecalis

The antimicrobial susceptibility test (AST) revealed that all the (*n* = 47) *E. faecalis* isolates in wild fish were sensitive to nitrofurantoin, norfloxacin, and fosfomycin. The numbers of isolates that showed resistance to ampicillin and erythromycin were 39 (82.98%) and 35 (74%), followed by linezolid and penicillin 33 (70%). Only a minority of *E. faecalis* showed resistance to ciprofloxacin, levofloxacin, and chloramphenicol, two (4%). In addition, a small portion, three (6%) total isolates, were found resistant to vancomycin. Four-fifths, 35 (74%), of the isolated *E. faecalis* from wild fish were found to be multidrug-resistant, with twelve resistant-patterns, where ten patterns were MDR and the MAR index ranged from 0.07 to 0.53 (Appendix A). The bivariate analysis revealed a significant correlation at the 0.01 level between erythromycin and linezolid (ρ = 0.00), penicillin and linezolid (ρ = 0.00), erythromycin and penicillin (ρ = 0.00), levofloxacin and ciprofloxacin (ρ = 0.00), vancomycin and ciprofloxacin (ρ = 0.00), vancomycin and levofloxacin (ρ = 0.009) (Appendix A). The study found no evidence of a significant association between antibiotics, different degrees of biofilm formation in *E. faecalis*, and the resistant gene in the *E. faecalis* in wild fish we used in this study (Appendix A).

The cultured fish samples showed a concerning trend of all (*n* = 40) *E. faecalis* isolates being resistant to linezolid, erythromycin, rifampicin, and penicillin. Furthermore, most of the isolates displayed resistance to ampicillin, 33 (82%), and fosfomycin and vancomycin, 3 (7%). A minority of *E. faecalis* isolates demonstrated intermediate susceptibility to ciprofloxacin, ampicillin, and vancomycin with numbers of six (15%), five (12%), and four (10%), respectively. Lastly, a small number of isolates, two (5%), exhibited intermediate resistance to chloramphenicol and fosfomycin. The cultured fish isolates were found to have MDR, with five resistance patterns observed and a MAR index ranging from 0.30 to 0.53 (Appendix A). Only fosfomycin and vancomycin significantly correlated in bivariate analysis (ρ = 0.00) (Appendix A). In contrast, crosstab analysis revealed a significant association between different degrees of biofilm formation by *E. faecalis* with fosfomycin, vancomycin, and the *bla_TEM_*-resistant gene. (Appendix A).

## 3. Discussion

### 3.1. Occurrence of Enterococcus faecalis

Enterococcus are most ubiquitously found in human and non-human (environments and animals) habitats [15]. It may cause human infections, especially in immunocompromised patients or weakened healthcare settings. *E. faecalis* is a leading cause of healthcare-associated infections, such as urinary tract infections, bacteremia, endocarditis, and surgical site infections [16]. According to CDC 2017, about fifty-five thousand cases related to Enterococci infections were reported in the United States, where five thousand four hundred people died [17]. Multidrug-resistant Enterococci are the primary concern of today’s world as they are resistant to most of the next-generation antibiotics that doctors have left available to prescribe in adverse situations. In the increasingly MDR *E. faecalis*, the biofilm-producing capability helps them combat antibiotics and potentially spread virulence genes via a horizontal gene transfer mechanism [18].

Until now, very little information has been published on isolating *E. faecalis* and comparing its prevalence in wild and cultured fish worldwide or in Bangladesh. The current study found a 54 and 46% detection rate of *E. faecalis* in wild and cultured fish, respectively. Elgohary et al. [19] studied *Oreochromis niloticus* fish. They found *E. faecalis* prevalence rates of 81.82% in EI Fayoum and 71.11% in EI Sharkia governates in Egypt, which are higher than the prevalence rate of *E. faecalis* in our study. Another study by Petersen and Dalsgaard [20] reported a 69% presence of *E. faecalis* in fish intestines from an integrated and traditional fish firm in Thailand. However, a recent study by Noroozi et al. [21] found a lower % prevalence of *E. faecalis* in fish, of 30%. In another study, Hassan et al. [22] isolated samples from cultured *Oreochromis niloticus* and *Mugil cephalus* in Egypt and found a lower prevalence of *E. faecalis* (50.50%) than ours. These variations might occur due to the variations in geographical and seasonal distribution, sample sizes, and types of methodology used for the whole procedure. To the best of our knowledge, this is probably the first study to investigate the variation of *E. faecalis* isolation from wild and cultured fish in Bangladesh, along with their biofilm-formation ability and virulence-gene profiling.

### 3.2. Antibiotic Resistance Pattern

Antimicrobial resistance (AMR) has emerged as a significant public health issue. Fish has an important impact on Bangladesh’s economy and cuisine, and it may be a reservoir for antibiotic-resistant *E. faecalis*, which can be transmitted to humans and animals at the market and during handling. In this study, *E. faecalis* isolated from wild and cultured fish intestines showed resistance against several classes of antimicrobials. Furthermore, our study found cultured fish more resistant than wild fish in the cases of the different antibiotics used in this experiment. *E. faecalis* from cultured fish exhibited higher resistance, e.g., more than 70% to the beta-lactam group of antibiotics, while lower resistance to tetracycline and vancomycin. All isolates from wild fish were susceptible to fosfomycin, a reserve group of antibiotics. Some previously reported studies from Bangladesh, Ghana, and the Persian Gulf also found similar results for these antibiotics [23,24,25]. In this study, in general, we found that *E. faecalis* strains from cultured fish showed higher levels of antibiotic resistance than those from wild fish, possibly due to the pressure of selective antibiotic uses in fish firms. Most fish farmers in Bangladesh utilize antibiotics throughout the entire cultivation cycle. Additionally, some employ antibiotics right from the start of production. This practice might contribute to the higher prevalence of antibiotic resistance in cultured fish compared to wild fish in the region [26].

A statistical analysis using a bivariate analysis was carried out for the resistant isolates in both types of samples. Antibiotic-resistant isolates from wild fish demonstrated highly significant correlations in the current study. A high-positive significant correlation was found between resistance patterns of penicillin and linezolid, erythromycin and penicillin, levofloxacin and ciprofloxacin, vancomycin and ciprofloxacin, and vancomycin and levofloxacin. However, the isolates from cultured fish exhibited a strong positive correlation only between fosfomycin and vancomycin (ρ = 0.00).

Multidrug-resistant (MDR) *E. faecalis* is a bacterium that is resistant to multiple antibiotics. The MDR *E. faecalis* has the potential to be a public health issue. Various foods and dietary sources may be a vehicle for multidrug-resistant *E. faecalis*. Finisterra et al. [27] reported that 70% of *E. faecalis* isolates collected from various food sources were multidrug-resistant. Most of the isolates analyzed in this study were found to be MDR. The resistance patterns found in *E. faecalis* from wild fish were more diverse (12 resistance patterns) than in *E. faecalis* isolates from cultured fish (five). Although insufficient data is available on the variation levels of MDR *E. faecalis* in wild and cultured fish, Arumugam et al. [28] reported that 62% of *E. faecalis* were found to be multidrug resistant. The variation in multidrug resistance in *E. faecalis* in wild and cultured fish, as well as high MDR *E. faecalis* in cultured fish, may be due to antibiotic overuse in the fish farm [29], with a dense fish population in a small area [30], mobile genetic-element transfer [31], and antibiotic exposure through feed [32]

### 3.3. Biofilm Formation and Virulence Genes

Biofilm or slime formation on media is linked to the formation of extracellular polysaccharides, which play an important role in bacterial adhesion. The crystal violet microtiter plate (CVMP) method revealed that the biofilm-producing ability in cultured fish samples was significantly higher than in wild fish samples. Approximately two-thirds of the isolates from wild fish and most isolates from cultured fish were biofilm formers to varying degrees. A previous study by Cui et al. [33] reported 16% of *E. faecalis* as strong biofilm producers and about 50% as intermediate biofilm producers. Another study by Toledo-Arana et al. [34] reported that 47% of *E. faecalis* isolates from different sources were strong biofilm producers on polystyrene plates. The discrepancy between the findings of these two test methods may be explained by the change in glucose concentration in the media employed in the current study. According to Rohde et al. [35], biofilm formation in the isolates may increase by up to 83% by adding 1% glucose to the TSB media.

The pathogenesis of *E. faecalis* depends on the bacteria’s ability to establish, adhere, invade, subvert the host’s defense mechanism, and form biofilms. This critical feature promotes the bacteria’s survival in harsh environmental conditions [36]. In this study, eight virulence genes, namely *agg*, *gelE*, *ace*, *pil*, *fsrA*, *fsrB*, *fsrC*, and *sprE*, but not *cyl*—which is responsible for host cell invasion and destruction [37]—were found in *E. faecalis*. Among these, virulence genes *agg*, *pil*, and *sprE* are related to *E. faecalis* biofilm formation [38]. The occurrence of these genes was higher in *E. faecalis* isolated from wild fish than in cultured fish. However, the important virulence genes in *E. faecalis* that directly contribute to biofilm formation and bacterial pathogenicity, *ace*, *gelE*, *fsrA*, and *fsrC,* were higher in number in cultured fish [39,40]. The statistical analysis revealed significant differences in biofilm-forming virulence genes between wild and cultured fish. This study clearly shows that *E. faecalis* has the potential to be a human pathogen, and *E. faecalis* from cultured fish showed more association with MDR and virulence factors.

The detection of biofilm-producing MDR *E. faecalis* in the cultured and wild fish is of public health concern because of their zoonotic potentiality. It also has clinical consequences. Directly or indirectly, there are possibilities for transmission of these bacteria to humans. In such cases, humans may become ill, and clinically, it may be difficult to treat such illnesses because of their MDR nature.

Nonetheless, there are some limitations to this study. Some statistical correlations were identified due to the small number of isolates, while others could have been resolved with larger isolates. Future research will address the current limitations and broaden the scope of the study by applying a whole genome sequencing-based approach to reveal the detailed resistomes and virulence gene distributions among isolates. The statistics will be revisited and expanded as more specimens become available to re-examine the findings with marginal or low significance.

## 4. Materials and Methods

### 4.1. Ethical Approval

The study procedures received approval from the Animal Welfare and Ethics Committee of Bangladesh Agricultural University (BAU) in Mymensingh, Bangladesh (AWEEC/BAU/2023(25)).

### 4.2. Sample Collection

From October 2021 to December 2022, a total of 60 koi fish (*Anabas scandens*), with an average weight of approximately 20 g and an average length of around 11 cm, as well as 72 stinging catfish (*Heteropneustes fossilis*), with an average weight of about 18 g and an average size of approximately 15 cm, were obtained from various local markets in the Mymensingh District of Bangladesh (located at 24.7500° N, 90.4167° E) (Figure 4). Among the 132 collected fish, 60 were wild (36 koi and 24 stinging catfish), while the remaining 72 were cultured fish (36 koi and 36 stinging catfish). To maintain aseptic conditions, all the samples were carefully collected in zip-lock bags and immediately transferred to the Department of Microbiology and Hygiene, Bangladesh Agricultural University, using an insulated icebox. The samples were taken from their intestines, homogenized in a mortar and pestle with phosphate buffer, and then transferred into the nutrient broth.

### 4.3. Isolation and Molecular Identification of E. faecalis

To enrich, approximately 1 mL of sample was placed in a 10 mL test tube with 5 mL of nutrient broth (HiMedia, Mumbai, India) and left overnight incubation at 37 °C in a shaker incubator with a rotation speed of 200 rpm. After enrichment, a loopful of sample was taken with a sterilized inoculum loop, streaked on Enterococcus Agar Base (EAB) media (HiMedia, India), and incubated at 37 °C for 24 to 36 h. The yellowish colonies on EAB media were suspected as being *E. faecalis* and confirmed by standard morphological (Gram-staining) and biochemical (sugar fermentation, catalase and Voges–Proskauer test) tests. Randomly selected isolates were subjected to MALDI-TOF [41] to identify *E. faecalis* at the species level.

For molecular identification of isolated *E. faecalis*, the most conserved *ddl_E. faecalis_* gene (F′-ATCAAGTACAGTTAGTCTT, R′-ACGATTCAAAGCTAACTG) [42] was detected by conventional PCR. The genomic DNA of the isolates for performing PCR was extracted, followed by the previously described boiling and chilling method [43]. In short, to extract the genomic DNA, 1 mL of an overnight broth culture was centrifuged at 5000 rpm for five minutes. The supernatant was discarded, and 500 μL of distilled water was added and, after vortexing, centrifuged again following the same parameters. After discarding the supernatant, 250 microliters of distilled water were added and vortexed. After the vortex, the isolates were boiled for 10 min and chilled for 10 min afterwards. Finally, samples were centrifuged at 10000 rpm for 10 min, the supernatant collected and stored at −20 °C until further use.

To perform PCR multiplication of the target gene, a 10µL of the final volume of PCR product was made where 10 µL of master mix, 2 µL double distilled water, 0.5 µL of forward and reverse primers, and 2 µL of genomic DNA were taken in a sterilized PCR nanotube and multiplied up to 30 cycles following the initial denaturation at 94 °C for 5 min (denaturation at 94 °C for 30 s, annealing at 50 °C for 90 s, and extension at 72 °C for 60 s) and a final extension at 72 °C for 10 min. After that, the amplified PCR products were subjected to electrophoresis, where the DNA products ran in a 1.5% gel with 100bp DNA marker at 100V electricity for 30 min and then were stained with ethidium bromide and visualized in a UV transilluminator (Biometra, Göttingen, Germany).

### 4.4. Antibiotic Resistance, Biofilm, and Virulence Factors

PCR-confirmed *E. faecalis* were then further investigated for the detection of virulence factors and antibiotic resistance. For this purpose, conventional PCR was employed, utilizing specific primers listed in Table 1 below, each associated with distinctive band sizes.

### 4.5. Phenotypic Antimicrobial Resistance

The antibiotic susceptibility test (AST) for PCR-positive *E. faecalis* was performed according to CLSI [46] guidelines by the previously described Kirby–Bauer disc diffusion method [47]. For the AST, 13 common and commercially available antibiotics under ten different classes were selected, including ansamycins (rifampin, 5 μg), fluoroquinolones (ciprofloxacin, 5 μg, levofloxacin, 5 μg, and norfloxacin, 10 μg), fosfomycin (fosfomycin, 50 μg), glycopeptides (vancomycin, 30 μg), macrolides (erythromycin, 15 μg), nitrofurans (nitrofurantoin, 300 μg), penicillins (penicillin, 10 μg and ampicillin, 10 μg), phenicols ((chloramphenicol, 30 μg), tetracyclines (tetracycline, 30 μg), and oxazolidinones (linezolid, 30 μg). According to the AWaRe classification of antibiotics by the World Health Organization, out of 13, 5 (ampicillin, chloramphenicol, nitrofurantoin, penicillin, and tetracycline) antibiotics were from the access group, 6 (ciprofloxacin, erythromycin, levofloxacin, norfloxacin, rifampin, and vancomycin) were from the watch group, and 2 (fosfomycin and linezolid) were from the reserve group. A pure colony from the EBA media plate was cultured overnight in a 1.5-microliter Eppendorf tube with nutrient broth. After 18–24 h of incubation, the cultured broth was poured into another Eppendorf tube and adjusted to 0.5 McFarland standard to perform the AST. Next, using a cotton swab, approximately 100 microliters of culture broth were taken by whirling and spread on Mueller–Hinton Agar (HiMedia, India) plates. Selected antibiotic disks were placed on the dispersed plates and incubated at 37 °C for 18 to 22 h. After incubation, isolates that exhibited resistance to three or more antibiotics were recorded as multidrug resistant (MDR) [48]. The following formula was used to calculate the multiple antibiotic resistance (MAR) index [49].
MAR = u/v
where u = number of antibiotics to which *E. faecalis* showed resistance and v = the total number of antibiotics that were used in the study.

### 4.6. Biofilm Formation Analysis

Biofilm forming capabilities of isolated *E. faecalis* were determined by crystal violet microtiter plate method using 96-well polyester microtiter plates following a previously described method [50]. In brief, freshly grown *E. faecalis* colonies on Congo Red Agar (CRA) (HiMedia, India) plates were transferred into 1 mL tryptic soya broth (TSB) (HiMedia, India) with 1% glucose supplement and incubated for 24 h at 37 °C without shaking. After that, the cultured broth was diluted into 1:100 in TSB, having 1% glucose. An amount of 200 µL was poured into the wells of 96 microtiter plates and incubated for 24 h (Figure 5).

A well of a microtiter plate containing only 200 µL of TSB with 1% glucose is considered a negative control for the test. After incubation, the planktonic bacteria in the microtiter plate were washed out three to four times with sterile phosphate buffered saline (PBS). After fixing the adherent bacteria to the well with 95% ethanol, the wells were stained with 1% crystal violet and air-dried before being washed with distilled water. Finally, the optical density (OD) value of every tested well was measured by a microtiter plate reader (VWR, part of Avantor, Radnor, PA, USA) at a wavelength of 570 nm. The isolates with OD_570_ nm value ≥ 1 were considered strong biofilm producers, 0.1 ≤ OD_570_ nm value < 1 as intermediate, and OD_570_ nm value < 0.1 considered non-biofilm former [51].

### 4.7. Statistical Analysis

Statistical analysis was conducted within different variables to see any variations and significance of the data. All the primary data from this study were entered into an Excel 2019 (Microsoft/Office 2019, Redmond, WA, USA) sheet for further analysis. Statistical Package for Social Science (SPSS.v.25, IBM, Chicago, IL, USA) and GraphPad Prism (Prism.v.8.4.2, San Diego, CA, USA) were also used to calculate the prevalence of different variables and to find any correlation between two of them. The chi-square test for relatedness analyzed variations between two variables in various degrees of occurrence with the help of the Z-test for proportion, where *p*-value ≤ 0.05 was the level of significance standard. Furthermore, the Pearson correlation coefficient analysis was conducted to see if there was any significant correlation between any two resistant antibiotics and between two virulence genes present in *E. faecalis*.

## 5. Conclusions

*E. faecalis* are opportunistic zoonotic pathogens. To the best of our knowledge, this is the first study in Bangladesh describing the molecular evidence-based detection of biofilm-producing MDR *E. faecalis* from wild and cultivated koi and catfish. The antibiotic-resistant *E. faecalis* detected in these fish has the potential to end up in the food chain to transmit to humans and other hosts. Horizontally, many of these resistances could also be transmitted to other bacterial species circulating in the same ecosystem. Therefore, the resistance pattern of these pathogens needs to be monitored from time to time to enable the use of effective antibiotics in aquaculture if required to reduce the burden of AMR on the ecosystem.

## Figures and Tables

**Figure 1 antibiotics-12-01375-f001:**
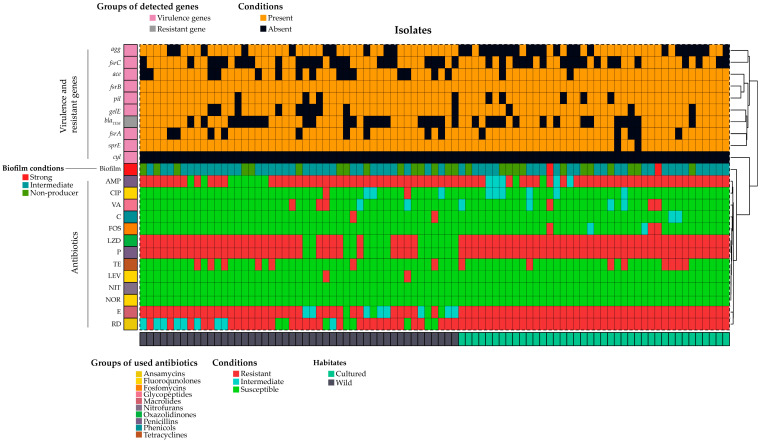
Heat map presentation of biofilm formation, virulence genes, and antibiotic resistance of isolated *E. faecalis*. Here, AMP = ampicillin; C = chloramphenicol; LZD = linezolid; CIP = ciprofloxacin; NIT = nitrofurantoin; E = erythromycin; FOS = fosfomycin; NOR = norfloxacin; RD = rifampicin; P = penicillin; TE = tetracycline; LEV = levofloxacin; VA = vancomycin.

**Figure 2 antibiotics-12-01375-f002:**
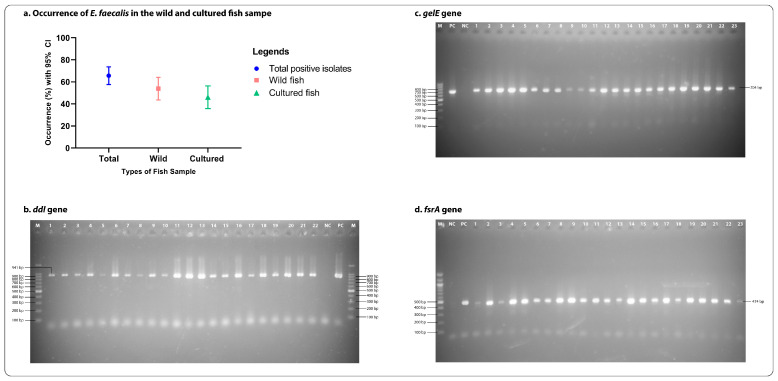
(**a**). Occurrence of *E. faecalis* in the samples, (**b**) virulence gene *ddl* (941 bp), (**c**) virulence gene *gelE* (704 bp), (**d**) virulence gene *fsrA* (474 bp).

**Figure 3 antibiotics-12-01375-f003:**
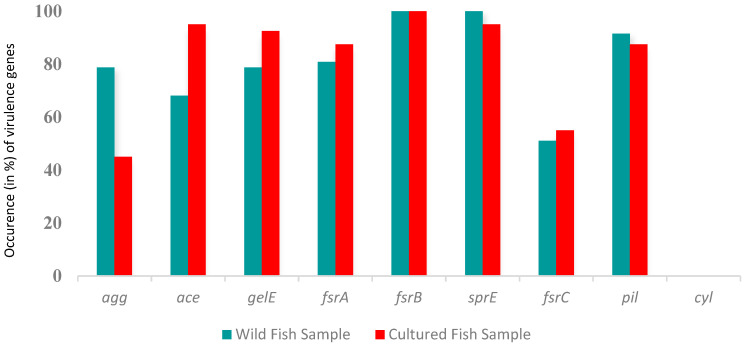
Variations of virulence genes distribution in the wild and cultured fish samples.

**Figure 4 antibiotics-12-01375-f004:**
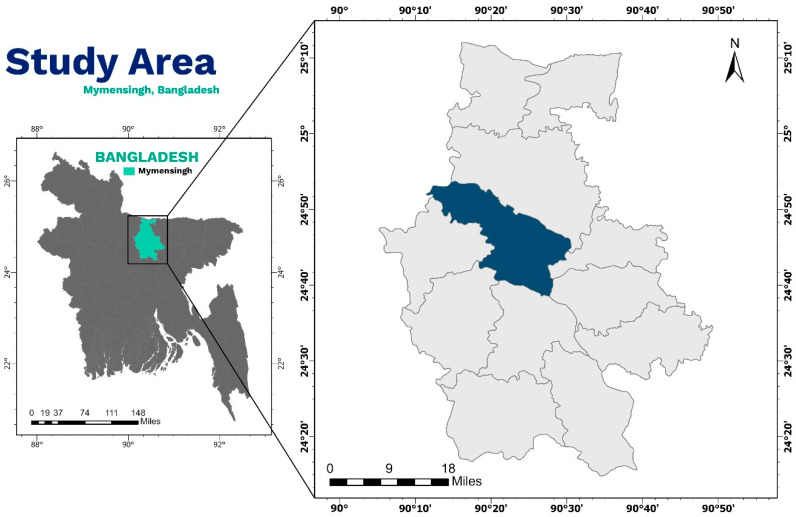
Study area of this study. Prepared using ArcGIS pro, ESRI, Redlands, CA, USA.

**Figure 5 antibiotics-12-01375-f005:**
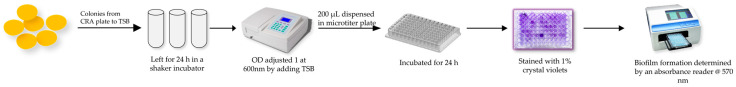
A flowchart of the crystal violet test method to determine biofilm formation.

**Table 1 antibiotics-12-01375-t001:** List of primers engaged with virulence and antibiotic resistance.

Factors	Target Genes	The Nucleotide Sequence (5′-3′)	Annealing Temperature	Amplicon Size (bp)	References
Species identification	*ddl*	F′-ATCAAGTACAGTTAGTCTT R′-ACGATTCAAAGCTAACTG	50	942	[42]
Biofilm	*agg*	F′-TCTTGGACACGACCCATGAT R′-AGAAAGAACATCACCACGAGC	58	413	[44]
Virulence	*gelE*	F′-GGTGAAGAAGTTACTCTGAC R′-GGTATTGAGTTATGAGGGGC	52	704	
	*ace*	F′-GAATGACCGAGAACGATGGC R′-CTTGATGTTGGCCTGCTTCC	58	615	
	*pil*	F′-GAAGAAACCAAAGCACCTAC R′-CTACCTAAGAAAAGAAACGCG	53	620	
	*fsrA*	F′-CGTTCCGTCTCTCATAGTTA R′-GCAGGATTTGAGGTTGCTAA	53	474	
	*fsrB*	F′-TAATCTAGGCTTAGTTCCCAC R′-CTAAATGGCTCTGTCGTCTAG	55	428	
	*fsrC*	F′-GTGTTTTTGATTTCGCCAGAGA R′-TATAACAATCCCCAACCGTG	54	716	
	*sprE*	F′-CTGAGGACAGAAGACAAGAAG R′-GGTTTTTCTCACCTGGATAG	53	432	
Antibiotic-Resistant gene	*Bla_TEM_*	F: CATTTCCGTGTCGCCCTTAT R: TCCATAGTTGCCTGACTCCC	56	793	[45]

## Data Availability

Not applicable.

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
