# Peer review of "Antimicrobial Resistance, Biofilm Formation, and Virulence Determinants in Enterococcus faecalis Isolated from Cultured and Wild Fish"

_antibiotics, 2023, doi:10.3390/antibiotics12091375_

Round 1

Reviewer 1 Report

This study characterized the E. faecalis isolated from cultured and wild fish as compare by AR profiles, biofilm formation, and virulence factors. There are some comments as follows;

1.     Re-calculate the proportions, 78.33% to 54% (L 91) and 55.56% to 46% (L 92)

2.     All data obtained from this study should be statistically analyze to determine the significance difference among fish samples at p<0.05.

3.     Fig. 2 – Heat map is not easy to see the difference. It should be displayed in Figure (box plot) or Table including LSD.  

4.     All isolates should be coded and analyze it to see the individual properties. It should be more informative to see the relationship among virulence, biofilm formation, and antibiotic resistance.

5.     The antibiotic resistance profiles between samples should be discussed in terms of horizontal gene transfer and acquisition of antibiotic resistance. For instance, the similarity in AR patterns between samples and relationship between antibiotic use and AR patterns should be more informative.

L 2: biofilm production to biofilm formation

L 11: potential source to potential reservoir

L 20: biofilm formers

L 23-24: multidrug resistant (MDR)

L 28: MDR (use abbreviation since mentioned earlier)

L 42: information to infection

L 47: those bacteria

L 50: USA

Extensive editing of English language required.

Reviewer 2 Report

Major issues

1. Objectives of the study must clearly described.

2. Please provide details about control isolates used for evaluation of the biofilm formation by the bacteria isolated in the study.

3. Please described criteria for declaring bacteria as strong, mild or no biofilm-forming. How did you deal with mild biofilm-forming isolates?

4. Please include a new, separate session to present all the associations between antibiotic resistance and biofilm-formation by the isolates obtained in the study.

Minor issues

1. Possibly, the Introduction can be divided into three separate sub-sections.

2. Instead of using Pearson correlation, I suggest to try Spearman correlation for more accurate results.

3. No need to express results with two decimal digits.

4. The Discussion can be divided into two or three sub-sections to facilitate reading.

5. Please include an overview of relevant findings from other parts of the world.

Reviewer 3 Report

Dear authors,

it is an interesting study on the AMR and virulence of Enterococcus in Bangladesh.

The authors use different terms to describe the sampled group of fish. In the abstract - local and cultured, inland culture and capture inlands, aquaculture in the introduction, then wild and cultured in the results and discussion. Please revise and use the same definitions throughout the manuscript.

The discussion needs improvement: authors mostly rewrite their results without any interesting linkage with the situation in Bangladesh. Could the authors provide some details on the consumption of antimicrobials in aquaculture in Bangladesh?

My other comment could be found in the attached file.

English revision is needed.

Round 2

Reviewer 1 Report

This has been well revised according to the reviewers' comments. 

Minor comment - In Figure 3, add the Y-axis title.

Minor editing of English language required

Author Response

Minor comment - In Figure 3, add the Y-axis title.

Response: Thank you so much for your valuable comment. In Figure 3, we have updated the figure with the Y-axis title and now available in the revised version of the manuscript.

Reviewer 2 Report

The authors have improved the manuscript during revision.

Before acceptance, they need to include a new paragraph describing the clinical consequences of their findings.

Then, the manuscript can be accepted.

Author Response

Before acceptance, they need to include a new paragraph describing the clinical consequences of their findings.

Then, the manuscript can be accepted.

Response: Thank you so much for your excellent comment on improving the manuscript. We have considered the suggestion and added a new paragraph on the clinical consequences of the current findings in the last part of the discussion before the limitation of this study. Please see the submitted revised version of the manuscript.

Reviewer 3 Report

Dear authors,

Thank you for addressing my comments on the manuscript.

Please find some of my additional comments below: 

Fig.1. Please add for b,c, and d at least a size of the genes of interest that will help a reader to understand your results. 

Fig.2. Could the authors consider adding %? Or at least horizontal gridlines?

Line 181. Please use Enterococcus or enterococci starting with lowercase. 

In the discussion, please add the number of isolates beside % where possible. 

Line 210. Enterococcus isolates of cultured fish?

Line 215. Results about resistance toward beta-lactam antibiotics? Please explain. 

Line 233. MDR in nature. Does that mean "shared intrinsic resistance"?

Minor language revision may be required.

Author Response

Dear authors,

Thank you for addressing my comments on the manuscript.

Response: We appreciate your time to review our manuscript.

Please find some of my additional comments below:

Fig.1. Please add for b,c, and d at least a size of the genes of interest that will help a reader to understand your results.

Response: Thank you so much for your nice suggestion. We have updated Figure 1 as suggested.

Fig.2. Could the authors consider adding %? Or at least horizontal gridlines?

Response: Thank you so much for your suggestion. The heatmap represents the overall occurrence of tested antibiotics and detected virulence genes in both wild and cultured fish samples. So, we -the authors- think this could not be appropriate to show the overall occurrence in % as it may convey a misinterpretation of our study. Furthermore, there are horizontal gridlines present in Figure 2. Although, we want to thank you for your nice suggestion and time to comment on the manuscript.

Line 181. Please use Enterococcus or enterococci starting with lowercase.

Response: Thank you so much for your comment. We have updated the Line 181.

In the discussion, please add the number of isolates beside % where possible.

Response: Thank you so much for your comment. We tried to add and update the numbers where applicable.

Line 210. Enterococcus isolates of cultured fish?

Response: Thank you so much for your valuable comment. We have updated the information.

Line 215. Results about resistance toward beta-lactam antibiotics? Please explain.

Response: Thank you so much for your comment. Its mentioned in result section. We tried to avoid the repetition of results in discussion and did not pout the results about resistance toward beta-lactam antibiotics here. The resistance was more than 70 toward beta-lactam antibiotics.  Now mentioned in line 215 as suggested.

Line 233. MDR in nature. Does that mean "shared intrinsic resistance"?

Response: Thank you so much for your valuable comment. We did not want to mean shared intrinsic resistance. We have updated the information appropriately.

Please see the submitted revised version of the manuscript.